# Sustainable Binary Blending for Low-Volume Roads—Reliability-Based Design Approach and Carbon Footprint Analysis

**DOI:** 10.3390/ma16052065

**Published:** 2023-03-02

**Authors:** Gudla Amulya, Arif Ali Baig Moghal, Abdullah Almajed

**Affiliations:** 1Department of Civil Engineering, National Institute of Technology Warangal, Warangal 506004, India; 2Department of Civil Engineering, College of Engineering, King Saud University, P.O. Box 800, Riyadh 11421, Saudi Arabia

**Keywords:** calcium lignosulfonate, CBR, carbon footprint analysis, clay, granite sand, reliability, subgrade

## Abstract

The utilization of industrial by-products as stabilizers is gaining attention from the sustainability perspective. Along these lines, granite sand (GS) and calcium lignosulfonate (CLS) are used as alternatives to traditional stabilizers for cohesive soil (clay). The unsoaked California Bearing Ratio (CBR) was taken as a performance indicator (as a subgrade material for low-volume roads). A series of tests were performed by varying the dosages of GS (30%, 40%, and 50%) and CLS (0.5%, 1%, 1.5%, and 2%) for different curing periods (0, 7, and 28 days). This study revealed that the optimal dosages of granite sand (GS) are 35%, 34%, 33%, and 32% for dosages of calcium lignosulfonate (CLS) of 0.5%, 1.0%, 1.5%, and 2.0%, respectively. These values are needed to maintain a reliability index greater than or equal to 3.0 when the coefficient of variation (COV) of the minimum specified value of the CBR is 20% for a 28-day curing period. The proposed RBDO (reliability-based design optimization) presents an optimal design methodology for designing low-volume roads when GS and CLS are blended for clay soils. The optimal mix, i.e., 70% clay blended with 30% GS and 0.5% CLS (exhibiting the highest CBR value) is considered an appropriate dosage for the pavement subgrade material. Carbon footprint analysis (CFA) was performed on a typical pavement section according to Indian Road Congress recommendations. It is observed that the use of GS and CLS as stabilizers of clay reduces the carbon energy by 97.52% and 98.53% over the traditional stabilizers lime and cement at 6% and 4% dosages, respectively.

## 1. Introduction

Weak soils are broadly distributed and pose different challenges to civil engineers. Such soils need to be improved internally to meet respective field requirements. The strengthening of soil layers for different applications in the geotechnical engineering domain is achieved by stabilization. The stabilization of soil is an effective method to improve its properties. It is subdivided into two categories: mechanical stabilization and chemical stabilization [1]. Chemical stabilization is a widely adopted method to enhance inferior soils. Traditional stabilizers such as lime [2,3,4], cement [5,6,7,8], and gypsum [9,10] have long been extremely common practices to treat weak soils. However, their treatments lead to a negative impact on the environment in terms of carbon emissions, high energy consumption for the production of materials, and changes in the pH of the soil. The sustainable use of resources that address the above challenges is a viable solution to protect the environment and conserve non-renewable resources [11]. In this context, researchers and engineers have introduced non-traditional stabilizers, claiming their sustainability, including biopolymers [12], microbially induced calcite precipitation (MICP) [13], polypropylene fiber [14], biochar [15], mine waste such as coal gangue [16,17], and industry by-products such as sandstone [18], marble dust [19], granite sand [20], limestone dust [21].

Industry by-products occupy a vast area because of massive dumping near work sites, which creates problems for humans and the environment. Using such materials in the construction industry resolves economic and environmental issues. Granite sand is one of the by-products obtained from the aggregate-crushing industry and has a high specific gravity. The massive production and deposition of granite sand create environmental issues [20]. Typically, in a year, 200 million tons of quarry waste is generated by the stone-crushing industry [22]. According to the Indian Bureau of Mines 2019, the total volume of granite produced is 61,16,085 cu·m. The amount of waste generated during the processing of granite stone is reported to be 50% of the finished granite product volume. The production of the finished stone after the quarry in 2015 was 82.6 million tons, while the waste generated during processing and extraction was about 70% of the total volume, and in 2022, the waste generated was 80% of the total volume [23]. These statistical data ensure that GS is a renewable material that can be used for various geotechnical applications.

GS is a non-plastic material with high shear strength and specific gravity [20,24]. Cheah et al. [24] used granite quarry dust (GQD) as a sand replacement at dosages up to 100% in intervals of 20% in a ternary blended cementitious composite. The composite material yielded better performance in flexural and compression analyses with up to a 60% replacement of GQD (granite quarry dust) due to the densification of the matrix. Sivrikaya et al. [25] worked on three different types of clays to improve Atterberg limits and compaction attributes. Different stone wastes, namely, marble dust of calcite and dolomite and granite powder, were used to enhance the soil properties. The effect of granite powder on the Atterberg limits and compaction attributes of high-plasticity clay was more significant than that of the other two materials. A clay soil was treated with granite dust in order to enhance its properties. It was observed that the CBR improved with a 20% addition of granite dust, which is due to agglomeration [26]. Sudhakar et al. [27] treated expansive clay with quarry dust to meet the foundation requirements of the pavement. The unconfined compressive strength of the soil yielded a higher value at a 15% addition of quarry dust due to bonding between the soil and cementitious compounds. This work also compared the thickness of the pavement with untreated and treated soils, where a 6 cm difference was observed under heavy traffic conditions. Okonkwo et al. [28] treated a lateritic soil with lime and quarry dust as subbase materials for low-cost roads. The work improved the strength and CBR performance by 230% and 126%, respectively, at 8% lime and 18% quarry dust.

Though there have been studies on using quarry dust, there is always the creeping point that the utilization of quarry dust requires its mechanical stabilization, which decreases the cohesion intercept due to the presence of sharp edges and the rough texture of the matrix [29]. This is addressed by some works, such as [20,29,30,31,32], where secondary additives were used. However, the soil becomes alkaline and brittle from pozzolanic reactions, which is unsuitable for the groundwater table and vegetation. To counteract this behavior of the soil, a suitable binder, such as lignin-based material, is required.

Lignin is a by-product in the form of a black liquid with high moisture content or a powder obtained from the biomass industry. The molecular weight and the chemical structure of lignosulfonate depend on the wood from which it is extracted [33]. Approximately, the production of paper waste every year is around 50 million tons [34]. Cheng et al. [35] stated that about 70 million tons of lignin is unused in the paper industry every year. Discharging these by-products leads to serious environmental issues. In this regard, the utilization of well-known lignin-based compounds such as calcium lignosulfonate (CLS) is considered an eco-friendly, low-carbon-emitting material and a savior of natural resources [36]. A comparative study was performed in the laboratory on untreated silty soil and silty soil treated with CLS. The work reported that CLS significantly improved the strength and CBR of the treated soil [37]. Indraratna et al. [38] treated a dispersive soil with CLS against erosion. The study concluded that it was a promising stabilizer for improving resistance to erosion and was a likely alternative to traditional stabilizers such as lime and cement. Koohpeyma et al. [39] investigated the potential of lignosulfonate treated with clayey sand to improve resistance to erosion. The work concluded that the addition of 4% lignosulfonate led to a significant improvement in erosion resistance. Sabitha and Sheela [40] studied the effect of CLS on the CBR behavior of Kuttanad clay. The study reported that the strength of the soil was increased with an increase in curing time.

Earlier works discussed the effects of GS and CLS individually on clay. However, no research has been performed exploring their combined performance on clay. This paper contributes to the field by examining the impacts of GS and CLS on clay. Two best-fit equations are presented for the estimation of the CBR of treated and untreated clay. This study may be the first to offer the optimal dosages of GS and CLS required to stabilize low-volume roads considering CBR strength failures by targeting the desired values of reliability indices as per codal provisions. This work was also extended by conducting a carbon footprint analysis (CFA) on the materials for pavement subgrade application. Carbon footprint analysis is a simplified procedure of Life Cycle Analysis (LCA). This process includes assessing CO_2_ emissions with or without greenhouse gases (GHGs), reducing complexities, and creating transparency. CFA has been adopted in various civil engineering works, such as pavement, embankments, and reinforced concrete structures. The application of CFA is limited in geotechnical engineering due to various soil profiles and designs. Ashfaq et al. [41], Vukotic et al. [42], Inui et al. [43], Shillaber et al. [44], Harmsel [45], and Bouazza and Heerten [46] performed works related to CFA in geotechnical engineering pertaining to structural elements, retaining wall structures, ground improvement methods, geotextile tubes, and geosynthetics, respectively.

In this study, the optimal mix was identified from the CBR results, and the appropriate dosage was selected as the subgrade material of flexible pavement. The C/S of the pavement was assumed based on the Indian Road Congress (IRC) recommendations.

## 2. Materials and Methods

### 2.1. Soil

The soil for this study was collected at a depth of 3ft from the ground surface at Battupally lake, Telangana, India (17.9737° N, 79.5352° E). The physical index properties and chemical composition are shown in Table 1. The soil was classified as intermediate compressible clay per the Unified Soil Classification System (USCS) performed according to ASTM-D2487 [47], with predominant silica and alumina present in the soil. The composition presented in Table 1 was obtained from XRF (X-ray Fluorescence) spectrometric analysis. Figure 1 displays a scanning electron microscopy (SEM) image of clay. The texture of clay particles is described at a magnification of 5 µm, where a honeycomb structure with several voids is observed.

### 2.2. Primary Additive

Granite sand (GS) was sourced from a quarry industry in the Gudipadu region of Telangana State, India, bearing the geographical coordinates 18.83793° N and 79.424954° E. It is an inert material produced by the primary crushing stage of aggregates. The physical index properties and chemical compound distribution are described in Table 1. GS is classified as poorly graded silty sand as per the Unified Classification (Unified Soil Classification System) performed according to ASTM D2487 [47]. GS was subjected to an electron beam generated by the scanning electron microscope to study the fabric of the particle. Figure 1 is a micrograph obtained from SEM analysis that shows that the particles of GS are angular–subangular, flaky, and completely granular.

### 2.3. Secondary Additive

Calcium lignosulfonate (CLS) was obtained from Aditya Chemicals, Hanamkonda region, Warangal, Telangana, India. It is an amorphous yellow-brown powder hydrophilic in nature and contains a benzene ring that is hydrophobic in nature, as shown in Figure 2. It comprises carbon, oxygen, sulfur, calcium, sodium, and potassium [48]. Generally, lignosulfonates are acidic in nature and soluble in H_2_O, though they do not dissolve in organic solvents [49]. CLS was examined for its physical and chemical properties. The color is observed as yellow-brown, and the pH is 4.3, which represents an acidic additive. The molar mass of CLS is 528.6 g/mol. The following mix proportions involving primary and secondary additives were adopted in the present study.

M1: 70% clay and 30% GSM2: 60% clay and 40% GSM3: 50% clay and 50% GSM1CLS0.5: 70% clay and 30% GS and 0.5% CLSM1CLS1: 70% clay and 30% GS and 1% CLSM1CLS1.5: 70% clay and 30% GS and 1.5% CLSM1CLS2: 70% clay and 30% GS and 2% CLSM2CLS0.5: 60% clay and 40% GS and 0.5% CLSM2CLS1: 60% clay and 40% GS and 1% CLSM2CLS1.5: 60% clay and 40% GS and 1.5% CLSM2CLS2: 60% clay and 40% GS and 2% CLSM3CLS0.5: 50% clay and 50% GS and 0.5% CLSM3CLS1: 50% clay and 50% GS and 1% CLSM3CLS1.5: 50% clay and 50% GS and 1.5% CLSM3CLS2: 50% clay and 50% GS and 2% CLS

### 2.4. California Bearing Ratio (CBR)

#### 2.4.1. Sample Preparation with GS

Samples for the CBR test (15 cm diameter and 17.5 cm height) were prepared by mixing GS with clay at replacement dosages of 30%, 40%, and 50% of the total mass of the soil. The mass of the soil was measured according to the Optimum Moisture Content and Maximum Dry Density of clay obtained in the Standard Proctor test according to ASTM-D698 [50]. Clay-GS samples were added to the mold and compacted into 5 layers and 56 blows each and tested for the CBR according to ASTM-D1883 [51].

#### 2.4.2. Sample Preparation with GS and CLS

Samples were prepared via the binary blending of GS and CLS with clay. For each clay-GS mix, 0.5%, 1%, 1.5%, or 2% CLS was added. The aliquot was obtained by mixing the measured quantity of CLS with the corresponding water content. Clay-GS-CLS samples were left undisturbed for mellowing in an airtight double-sealed plastic bag for 24 h [52]. Each clay-GS-CLS sample was cured by placing the sample under a double zip-locked cover for 0, 7, and 28 days before it was tested.

### 2.5. Microstructural Analysis

#### 2.5.1. Scanning Electron Microscopy Analysis (SEM)

Scanning electron microscopy (SEM) tests were performed to examine the surface morphological changes in the GS-CLS-treated clay. TESCAN VEGA 3LMU microscopy with a heated tungsten cathode and 3D beam technology was used in this study. A very fine oven-dried sample of around 2 mg was coated to avoid the charring effect of the splutter coater before it was studied under the microscope. The micrographs depict the inherent mechanism of treated samples. The magnified micrographs that best illustrate the peculiar microstructure of the soil are displayed in a later section.

#### 2.5.2. Fourier Transform Infrared Spectroscopy (FTIR)

FTIR analysis was carried out using Perkin Elmer 100 s (Spectralab Scientific Inc., Markham, ON, Canada) to identify the functional groups present in the studied soil, GS, and CLS. Infrared spectra were recorded using a standard DTGS detector in the spectral range of 350–4000 cm^−1^ at a resolution of 4 cm^−1^ with a KBr (Potassium bromide) pellet arrangement. The outcome of the analysis is in the form of a spectrum that indicates the bonds between atoms.

### 2.6. Low-Volume Roads (LVRs)

Around 70% of the world’s roads comprise low-volume roads. These are the roadways that carry traffic volumes with less than 450 commercial vehicles per day (CVPD). These require huge amounts of maintenance. These are certainly unpaved roads with 400 vehicles/day. The most exposed layer is typically affected by material deformation and dust erosion. Materials that exhibit a ductile nature are less susceptible to damage created by traffic [53]. Though the traffic volumes on these roads are relatively less when compared to highways, they constitute a greater road length worldwide. Unpaved roads are highly susceptible to extreme weather conditions that lead to dust emission. The techniques used for the design of LVRs constantly change with respect to the existing circumstances. A treated LVR or sealed LVR is required to protect against material deformation and dust emission. Calcium chloride or magnesium chloride treatments are the most frequently adopted surface treatments for dust control. However, a thin asphalt layer is adopted for its better binding nature but is less effective [35]. In this context, the works by Cheng et al. [35] and Gafoori et al. [54] suggest that a stabilized material is more effective than a treated LVR or sealed LVR in realizing strength, resistance to material deformation, and dust control.

The current study proposes sustainable materials (granite sand and calcium lignosulfonate) with lower CO_2_ emissions as a dust palliative and to maintain the economy-and-ecology balance [20,24,42,55].

According to IRC 89:2010 [56], the stabilized CBR must be 15%. The addition of CLS to clay-GS satisfied the criteria for flexible pavement for a curing period of 28 days, which is reported in the subsequent sections. The CBR is higher for GS-CLS-stabilized clay. The CBR attained at 0 days of the curing period was >4%, and after 28 days of curing, it was enhanced to >20% for all clay-GS mixes with 0.5% CLS. According to IRC SP 72-2007 [57], the stabilized soil satisfied the subgrade strength criteria for low-volume roads.

### 2.7. Reliability Analysis

#### 2.7.1. Need for Reliability-Based Design

Most of the existing village roads in India are unpaved low-volume roads. Low-volume roads fulfill a critical function where agriculture is the dominant economic activity. These roads provide access to agricultural communities for the mobility of people and the movement of goods from agricultural fields to markets. Rural roads accommodate a low volume of traffic with light transport vehicles. The frequency of heavy traffic is low. Therefore, low-volume roads are essential for socio-economic growth and the development of rural livelihoods. Pavements that are constructed with materials of marginal quality and that carry low levels of traffic have a low risk of pavement failure due to traffic loading. The suggested methods involved in designing low-volume roads are different from the traditional highway engineering standards.

It is noted from previous research that the design criteria for low-volume roads are considerably relaxed to achieve significant cost savings. Ultimately, this results in passenger discomfort. The poor condition of roads can be attributed to poor maintenance, harsh climatic conditions, unexpected heavy-traffic loading, and poor materials used in the construction due to the non-availability of good-quality soil. Therefore, many of the earlier approaches to the design of low-volume roads are inappropriate. Moreover, there is growing pressure for the construction of sustainable roads to replace good-quality materials with industrial by-products. Hence, there is a need to revise conventional design approaches considering the significant increases in knowledge, technology, and research. This has triggered the need for the development of new design guidelines for low-volume roads.

The proposed optimal mix for rural roads in India was defined based on the CBR. The overall assessment of low-volume roads depends on the variability in the materials tested and the equipment used for testing. The variability associated with the CBR value of low-volume roads can be attributed to heavy rainfall, low subgrade strength due to the use of marginal-quality materials, inadequate compaction, and frequent floods. The variability associated with the CBR of locally available clayey soil is also dependent on the inherent characteristics of the soil, method of sampling, method of testing, and field moisture content. Consequently, variability in the performance of low-volume roads is expected due to load and traffic analysis, environmental factors, and the evaluation of materials. The performance of these roads is greatly affected by the natural variations in the subgrade CBR. Early failures in low-volume roads may be expected due to this variability. Therefore, the variability in the design of low-volume roads is inevitable and must be taken into account in the design.

Conventionally, deterministic procedures were used for low-volume roads considering the limits of the design constraints. However, there is no scope to accommodate the variability. The optimal pavement thickness for low-volume roads obtained using deterministic optimization may have a high chance of failure when the uncertainties of subgrade strength (i.e., CBR) are not considered. The random distribution of GS and CLS within the clay may also induce uncertainties in CBR values. Reliability-based design optimization (RBDO) can be used to account for uncertainties associated with the design of low-volume roads. The design of low-volume roads using RBDO may yield safe pavement sections.

The target RBDO methodology developed by Basha and Babu [58,59,60,61] for the design of reinforced soil structures and cantilever sheet pile walls was used in the present investigation.

#### 2.7.2. Previous Studies on Reliability Analysis of Pavements

Divinsky et al. [62] reported the reliability analysis of pavements using CBR values. Kim and Buch [63] and Retherferd and McDonald [64] used point estimate methods (PEM), the first-order second moment (FOSM), and the first-order reliability method (FORM) to perform reliability analyses. Sani, Bello, and Nwadiogbu [65] presented a FORM for the design of pavements using the unconfined compressive strength and CBR. Moghal et al. [14,66] reported the RBDO procedure to study the effect of fiber reinforcement on the hydraulic conductivity and UCS of lime-treated expansive soils simultaneously. It is clearly noted from the review of the literature that the proposed recommendations are not appropriate in the harmonization of codes for the design of low-volume roads, as there are no standard guidelines available.

This is potentially the first study to propose the RBDO of low-volume roads by treating the CBR of the subgrade material and the dosages of GS and CLS of treated clay soils as random variables. The methodology developed in this paper to design low-volume roads takes care of permanent deformation. A multi-objective probabilistic optimization framework is proposed to select a clay-GS-CLS mixture such that the performance of the low-volume road is satisfactory against CBR failure. The proposed framework presented in this paper can be used to build low-volume roads with the selected clay by-product mixtures.

#### 2.7.3. Reliability Analysis Procedure

Reliability analysis was conducted to validate the experimental data. Regression equations were developed as the first step to correlate the CBR values with the dosage of GS, the dosage of CLS, and the curing period (CP). Then, the performance functions were developed using codal provisions to estimate reliability indices for the design of low-volume roads.

#### 2.7.4. Regression Analysis of the CBR Data Obtained from Experiments

The effects of the dosages of granite sand (GS) and calcium lignosulfonate (CLS) on the experimental values of the CBR obtained after 7- and 28-day curing periods are expressed through a linear regression equation. The dosages of GS and CLS and the curing time are independent variables, and the CBR is considered the dependent variable. The obtained CBR values are best represented by a linear equation [67]. This equation yields relatively good estimates of CBR with 24 data points. The equation to estimate the CBR of treated clay is given by
CBR_fit_ = a × D_GS_ + b × D_CLS_ + c × CP + d(1)
where, a, b, c, and d represent the regression coefficients, and D_GS_ and D_CLS_ represent the dosages of granite sand (GS) and calcium lignosulfonate (CLS), respectively. The equation proposed for the CBR of treated clayey soil can be written as
CBR_fit_ = −0.217 × D_GS_ − 0.363 × D_CLS_ + 0.415 × CP + 14.032 with R^2^ = 0.830(2)

Similarly, the equation proposed for the CBR of untreated clay is given by
CBR_fit_ = log (16.233 + 0.354 × CP) with R^2^ = 0.998(3)

Table 2 and Table 3 provide details of the regression analysis. Table 2 depicts the regression analysis for the CBR (CBR_fit_) of untreated clay measured after 0, 7, and 28 days of curing. In addition, the regression analysis of the CBR (CBR_fit_) of soil treated with granite sand (GS) and calcium lignosulfonate (CLS) measured after 7 and 28 days of curing is presented in Table 3. The coefficient of determination (R^2^) values for the CBR_fit_ of untreated and treated clayey soils are 0.830 and 0.998, respectively, as shown in Equations (2) and (3). This indicates that the experimental data are well predicted by the proposed regression equations.

#### 2.7.5. Performance Function for RBDO

The safety of low-volume roads constructed using treated clay may be expected when the CBR of clay is greater than or equal to the minimum specified CBR value (CBR_min_) [68]. The probabilistic measure for reliability can be defined as
Reliability= P (CBR ≥ CBR_min_)(4)

Subgrade failures of low-volume roads may be expected when the CBR of treated clay is less than the minimum specified CBR value (CBR_min_). Then, the performance function for the low-volume road failure is written as:g(x) = [CBR_fit_ − (CBR_fit_)/(CBR_min_)]_min_(5)

#### 2.7.6. Estimation of Reliability Indices Using FORM

The random space (X-space) is transformed into the standard normal random space (U-space) to perform nonlinear constrained optimization. The most probable point in the performance function is referred to as the design point (u*). The transformation between X = (D_GS_, D_CLS_, CBR_min_) and $U=\left\{\frac{D_{\mathrm{GS}}-\mu_{D_{\mathrm{GS}}}}{\sigma_{D_{\mathrm{GS}}}},\frac{D_{\mathrm{CLS}}-\mu_{D_{\mathrm{CLS}}}}{\sigma_{D_{\mathrm{CLS}}}},\frac{{\mathrm{CBR}}_{\min}-\mu_{{\mathrm{CBR}}_{\mathrm{mis}}}}{\sigma_{{\mathrm{CBR}}_{\min}}}\right\}$ is carried out at the design point (u*) [60]. The parameters μ(D_GS_), μ(D_CLS_), and μ(CBR_min_) represent the mean values of D_GS_, D_CLS_, and CBR_min_, respectively. The parameters σD_GS_, σD_CLS_, and σCBR_min_ represent the standard deviations of D_GS_, D_CLS_, and CBR_min_, respectively. The reliability index can be computed as follows:(6)$$\mathrm{Find}\;\beta_{\mathrm{CBR}}\;\mathrm{that}\;(a)\;\mathrm{minimizes}\;\surd (u^{T}u)\;\mathrm{or}\;(b)\;\mathrm{is}\;\mathrm{subject}\;\mathrm{to}\;g(u)\;\mathrm{where}\;\mathrm{either}\;(a)\;\mathrm{or}\;(b)\;\mathrm{is}\;0$$
where g(u) is the performance function against the CBR strength failure of low-volume roads in the U-space. The reliability index (β_CBR_) is given by
(7)$$\beta_{\mathrm{CBR}}=-\frac{\left(\begin{array}{c} \sigma_{D_{\mathrm{GS}}}\frac{\partial g}{\partial D_{\mathrm{GS}}}\left\{\frac{D_{\mathrm{GS}}-\mu_{D_{\mathrm{GS}}}}{\sigma_{D_{\mathrm{GS}}}}\right\}+\sigma_{D_{\mathrm{CLS}}}\frac{\partial g}{\partial D_{\mathrm{CLS}}}\left\{\frac{D_{\mathrm{CLS}}-\mu_{D_{\mathrm{CLS}}}}{\sigma_{D_{\mathrm{CLS}}}}\right\})\\ +\sigma_{{\mathrm{CBR}}_{\min}}\frac{\partial g}{\partial \left({\mathrm{CBR}}_{\min}\right)}\left\{\frac{{\mathrm{CBR}}_{\min}-\mu_{{\mathrm{CBR}}_{\min}}}{\sigma_{{\mathrm{CBR}}_{\min}}}\right\} \end{array}\right)}{\sqrt{{\left\{\sigma_{D_{\mathrm{CS}}}\frac{\partial g}{\partial D_{\mathrm{GS}}}\right\}}^{2}+{{\left\{\sigma_{D_{\mathrm{Cs}}}\frac{\partial g}{\partial D_{\mathrm{CLS}}}\right\}}^{2}+\{\sigma_{{\mathrm{CBR}}_{\min}}\frac{\partial g}{\partial \left({\mathrm{CBR}}_{\min}\right)}\}}^{2}}}$$

Refer to Basha and Babu [60] for the step-by-step procedure to determine $\beta_{\mathrm{CBR}}$. Furthermore, the mean values of D_GS_, D_CLS_, and CBR_min_ at the design point for the target value of β_CBR_ are expressed as
(8)$$D_{\mathrm{GS}}=\mu_{D_{\mathrm{GS}}}-\sigma_{D_{\mathrm{CLS}}}\beta_{\mathrm{CBR}}\left(\frac{\frac{\partial g}{\partial D_{\mathrm{CLS}}}\sigma_{D_{\mathrm{CLS}}}}{\sqrt{{\left\{\sigma_{D_{\mathrm{GS}}}\frac{\partial g}{\partial D_{\mathrm{GS}}}\right\}}^{2}+{\left\{\sigma_{D_{\mathrm{CLS}}}\frac{\partial g}{\partial D_{\mathrm{CLS}}}\right\}}^{2}+{\left\{\sigma_{{\mathrm{CBR}}_{\min}}\frac{\partial g}{\partial \left({\mathrm{CBR}}_{\min}\right)}\right\}}^{2}}}\right)$$
(9)$$D_{\mathrm{CLS}}=\mu_{D_{\mathrm{CLS}}}-\sigma_{D_{\mathrm{CLS}}}\beta_{\mathrm{CBR}}\left(\frac{\frac{\partial g}{\partial D_{\mathrm{CLS}}}\sigma_{D_{\mathrm{CLS}}}}{\sqrt{{\left\{\sigma_{D_{\mathrm{GS}}}\frac{\partial g}{\partial D_{\mathrm{GS}}}\right\}}^{2}+{\left\{\sigma_{D_{\mathrm{CLS}}}\frac{\partial g}{\partial D_{\mathrm{CLS}}}\right\}}^{2}+{\left\{\sigma_{{\mathrm{CBR}}_{\min}}\frac{\partial g}{\partial \left({\mathrm{CBR}}_{\min}\right)}\right\}}^{2}}}\right)$$
(10)$${\mathrm{CBR}}_{\min}=\mu_{{\mathrm{CBR}}_{\min}}-\sigma_{{\mathrm{CBR}}_{\min}}\beta_{\mathrm{CBR}}\left(\frac{\frac{\partial g}{\partial \left({\mathrm{CBR}}_{\min}\right)}\sigma_{{\mathrm{CBR}}_{\min}}}{\sqrt{{\left\{\sigma_{D_{\mathrm{CS}}}\frac{\partial g}{\partial D_{\mathrm{GS}}}\right\}}^{2}+{\left\{\sigma_{D_{\mathrm{Cs}}}\frac{\partial g}{\partial D_{\mathrm{CLS}}}\right\}}^{2}+{\left\{\sigma_{{\mathrm{CBR}}_{\min}}\frac{\partial g}{\partial \left({\mathrm{CBR}}_{\min}\right)}\right\}}^{2}}}\right)$$

Schaefer, White, Ceylan, and Stevens [68] presented guidelines for the design and construction of the subgrade and subbase layers of pavement systems for low-volume traffic. They reported that the minimum value of the CBR of the subgrade layer should be 10% to prevent pavement deterioration under traffic loadings. Therefore, the mean value of the minimum specified CBR (CBR_min_) is considered 10% in the present study for the design of low-volume roads.

The range of parameters considered for the RBDO of low-volume roads is shown in Table 3. The coefficient of variation (COV) of the dosage of GS (D_GS_) and the dosage of CLS (D_CLS_) is considered to be 5%, as these are controlled parameters. The CBR_min_ strengths are lognormally distributed with a maximum standard deviation of 0.6 times CBR_min_ [67]. The effects of adding granite sand (GS) and calcium lignosulfonate (CLS) to clay on the β_CBR_ are discussed.

### 2.8. Carbon Footprint Analysis (CFA)

Carbon footprint analysis is a process that analyzes the impact of a product or a material on the environment [68]. It is defined as all CO_2_ (direct) and methane (CH4) (indirect) emissions from any work/item and is reported in the form of equivalent CO_2_ emissions (e CO_2_) in terms of the Global Warming Potential (GWP). The constituents and levels of Global Warming gases change frequently, but CO_2_ emissions and carbon are the major factors among all aspects that create an impact on the environment. Hence, the CO_2_ emissions are considered in performing CFA [41,69,70,71]. This work involved the evaluation of CO_2_ emissions at different stages of pavement construction. A typical low-volume pavement subgrade is assumed based on IRC SP 72-2007 [57]. It is proposed that the subgrade be constructed with the stabilized clay suggested in the current study. The CO_2_ emissions are estimated during the construction of the subgrade.

To evaluate CO_2_ emissions, the procedure adopted is based on the approach followed by Ashfaq et al. [41,69], Shillaber et al. [71], and Hughes et al. [72]. The following are the steps involved in determining the CO_2_ emissions during the construction of the subgrade.

Stage I: Estimate the amount of carbon evolved from materials used for the pavement subgrade application.Stage II: Estimate the amount of carbon evolved during the procurement and haulage of the materials.Stage III: Estimate the carbon emissions during site operations.

In each stage, the carbon emissions of the material are calculated by considering the measured mass of soil and embodied carbon factor (ECF), which is obtained according to Hammond and Jones [73]. The ECF value obtained is free of greenhouse gases, which prevents variations and complexities.

#### Boundary Conditions

The soil used for the selected section must be the optimum among all other combinations. The best-performing material is 70% clay with 30% granite sand and 0.5% calcium lignosulfonate (M1CLS0.5). The testing method is performed by considering a uniform density and moisture content for all combinations. Hence, the following are the conditions adopted for calculating the carbon emissions:Equivalent carbon emissions are calculated based on the dosage of clay, GS and CLS.A uniform density of 1.75 g/cc is maintained to compact the soil for the entire section.A measurable moisture content of 16.3% is considered for effective compaction throughout the project.The manufacturing process is excluded from the calculations, as the materials are applicable for various purposes.The embodied carbon factor for maintenance and disposal processes is not considered because the selected material satisfied the technical requirement.

## 3. Results and Discussion

### 3.1. Variation in CBR with GS

The effect of the dosage of GS on clay was explored with the CBR test in this study. The clay-GS samples were tested for the CBR at a constant strain rate of 1.2 mm/min to determine the force required to reach 7 mm penetration, in accordance with ASTM D1883-21 [51]. The unsoaked CBR of the soil decreases with an increase in the dosage of GS, as depicted in Figure 3. This is ascribed to the change in the particle size distribution with the addition of an inert/coarse material [31]. The soil structure changes to the dispersed state, as shown in Figure 4. At a constant density and water content, an increase in the coarser fraction in clay will allow soil particles to slip over one another, which offers less resistance to the applied load. As shown in Figure 1, the voids in clay are filled with GS and may undergo mechanical stabilization. At modified compaction energy, the particles exhibit a dispersive nature, which fails to resist the static load. Hence, the recorded CBR value is lower than that of the virgin soil (Figure 3 and Figure 4).

### 3.2. Variation in CBR with GS and CLS

The CBR behavior of clay was enhanced with the curing period due to polymer chain formation. Several factors, such as silica, the predominant clay mineral, calcium, and the fineness of the soil, play a crucial role in polymer formation. Cation exchange, hydrogen bonding, and covalent bonding between compounds of clay-GS and CLS are responsible for polymer chain formation [52]. The effects of GS and CLS dosages and the curing period on the CBR of clay are interpreted in the subsequent sections.

#### 3.2.1. Effect of GS

With an increase in the dosage of GS, the CBR of blended clay decreased due to increase in the coarser fraction for every clay-CLS mix (M1 < M2 < M3). CLS bonding with clay and GS particles made the matrix show greater resistance to penetration when compared to clay. CLS underwent basal bonding with expansive minerals in the clay and peripheral bonding with non-expansive minerals in the clay and GS [52,55]. Due to weak bonding forces that act peripherally, an increase in the GS dosage yielded lower values, which is evidenced in Figure 5, Figure 6, Figure 7 and Figure 8 (Figure 5 through Figure 8). At a constant dosage of CLS, the soil changes from rounded, semi-angular, and angular flocs formed by the polymer action. The load corresponding to the depth of penetration for different GS contents was examined for 0-, 7-, and 28-day curing periods, as shown in Figure 5 through Figure 7. However, the CBR of the soil certainly decreases with an increase in the depth of penetration.

The behavior of the soil varied with different GS dosages. However, soil with 40% GS and 1.5% CLS behaves more elastically when compared to the other mixes but could not take higher loads. Soil with 40% GS and 2% CLS slipped into its plastic state at early strain levels but possessed a more ductile nature than all other mixes when cured for 28 days. The effect of variation in the GS content on the load–penetration behavior is very insignificant in the presence of CLS for any curing period. However, under all varying conditions, samples mixed with 50% GS (M3) yielded lower loads and strain and possessed low ductility, while those with 30% GS (M1) with CLS 1.5% take higher loads at shallower penetrations. Shah et al. [74] worked with lime, granite powder, and rhyolite to stabilize the clay. The CBR of the soil was enhanced by 166% due to the formation of cementitious compounds by lime, and the presence of high-specific-gravity material with the required quartz-to-feldspar ratio represents rhyolite and granite dust, which is in contradiction with the existing work. However, the presence of lime imparts a brittle nature to the soil, which may not be suitable for wheel loads. A silty clay soil was treated with calcium carbide residue (CCR) and granite dust to improve its penetration resistance. The treated soil yielded a higher response at 10% granite dust and 10% CCR, which was ascribed to the formation of chemical bonds [20]. Despite low-carbon-emitting materials, the development of pozzolanic reactions in the presence of CCR changes the pH of the soil, which becomes brittle for longer durations.

#### 3.2.2. Effect of CLS

The CBR of clay is highly influenced by the CLS dosage (Figure 8). At an initial dosage of CLS (i.e., 0.5%), the CBR of the soil is increased due to sufficient polymer chain formation irrespective of the GS dosage, as observed in Figure 5 through Figure 8 (M1CLS0.5, M2CLS0.5, and M3CLS0.5). Additionally, the presence of CLS improves the stability of clay through its dispersive action [75]. For dosages of 1% and 1.5%, there was a decrease observed in the CBR due to excess CLS due to the replacement of the soil with finer lignosulfonate (Figure 9) [40]. At a 2% dosage of CLS, for all GS mixes, the CBR of the soil is again improved but at lower increment rate (Figure 8). The excess CLS forms a filmy layer on the particles, which creates a pulling force on the particles, as depicted in Figure 9 (M1CLS2, M2CLS2, and M3CLS2). Load–penetration curves are plotted to observe the behavior of the CBR of treated soil in the presence of CLS for the curing period. The presence of CLS imparts a ductile nature to the soil [76]. However, an excess amount of lignosulfonate affects the workability of the mix due to the formation of finer lignosulfonate [40]. For all curing periods, at any dosage of GS, the soil mixed with 0.5% CLS takes higher loads at higher penetration values, as illustrated in Figure 5 through Figure 7. The soil exhibited a sufficient ductile nature, but the elastic nature of the soil lasted for a shorter period. The effect of CLS on the load–penetration behavior is not pronounced due to insignificant variations. Mudgal et al. [30] used lime and stone dust to enhance the properties of black cotton soil. The CBR was recorded as 22% for 9% lime and 20% stone dust due to pozzolanic reactions. Though the presence of calcium-based additives shows a rapid increase in engineering properties, it typically results in a high pH value, brittle failure, and stiff flocs. In contrast to this scenario, the usage of non-traditional additives leaves the soil without altering the pH of the groundwater and aids in long-term stability.

#### 3.2.3. Effect of Curing

With an increase in the curing period, the bearing resistance of the soil increase. In the early days of curing, the CBR of the soil is closer to the CBR of untreated clay, which is contradictory to the observations of Ta’negonbadi and Noorzad [77]. This rate of increase in strength is reduced due to the decreased formation of flocs. With the increase in the curing period, CLS neutralizes the negative charge and reduces the crystalline size of the clay mineral, which aids in stable aggregation (Figure 10) [78]. However, GS particles are inert and less reactive to the curing time, but they form a high-specific-gravity material, as explained in Table 1, which contributes to an increase in the bearing resistance. From Figure 5 through Figure 8, there is a significant rate of increase in the CBR for 28 days when compared to 0 and 7 days of curing, which is due to its slow reactivity. Figure 5 through Figure 7 present the load–penetration curves for 0-, 7-, and 28-day curing periods. It is noticeable that soils of all mixes tested after a longer curing period bear greater loads at initial penetrations when compared to the earlier curing periods. Additionally, it is possible that an increase in the curing period increased the load-carrying range (i.e., for 0 days, the range of the load is 30 kg–230 kg; for 7 days, the load range is 30–270 kg; and for 28 days, the load range is 30–470 kg). This improvement is ascribed to the formation of stronger bonds with time, as depicted in Figure 10 [79].

#### 3.2.4. Effect of Porosity 

The CBR of the treated soil is also influenced by its physical properties. The treated soil develops flocs as a result of aggregation, which leaves large pores that are fewer in number (Figure 9). These large pore spaces are fewer when compared to the small pore spaces present in the virgin soil, as observed in Figure 1 and Figure 10 (M1CLS0.5, M2CLS0.5, and M3CLS0.5), respectively. Changes in the porosity of the treated soil restricts the flow of water, and the presence of adsorbed CLS decreases the water-absorbing capacity. This helps in the further confinement of the soil and leads to an increase in the bearing resistance due to the optimized pore distribution [48,80]

#### 3.2.5. Fourier Transform Infrared Spectroscopic (FTIR) Behavior

The FTIR spectra of soil, GS, and CLS are presented in Figure 11, Figure 12 and Figure 13. In clay, constant stretching is observed at wavelengths of 800 cm^−1^ and 1055 cm^−1^, which is due to the Si-O bond, with lower intensities. Similarly, Al-O-H stretching is observed at a wavelength of 920 cm^−1^, and O-H molecular stretching is assigned to a wavelength of 3350 cm^−1^ (Figure 11). A similar spectrum was observed in the case of Panda et al. (2010), where Kaolinite clay was treated with sulfuric acid. In Figure 12, two bands are observed at 3430 cm^−1^ and 1640 cm^−1^, which is due to the -OH stretching and -OH deformation of water. A stretching band at 1567 cm^−1^ and a less intense band at 1015 cm^−1^ are observed due to Si-O stretching. A small Al-O-Si stretching band at 537 cm^−1^ and a band at 472 cm^−1^ are caused by the Si-O-Si bond. A similar spectrum was observed by Passaretti et al. [81], where biocomposites were derived from granite sand. Friedrich et al. [82] studied Muscovite, where a similar distribution of chemical bonds appeared. In Figure 13, in-plane bands are observed at wavelengths of 1500 cm^−1^, 1520 cm^−1^, and 1645 cm^−1^, which is due to methyne formation (C-H bond). A C-O bond (Alcohol) is observed at a wavelength of 1100 cm^−1^, which yields a sharp band. A band at a wavelength of 2713 cm^−1^ is observed due to the presence of sulfonate (S-O bond). These formations of an uninterrupted stretch of wavelengths confirmed that the fraction belongs to calcium lignosulfonate.

The CBR behavior of the soil is also evidenced by the FTIR behavior of the soil. The minute, thin polymer chains observed in Figure 10 are due to the formation of new chemical bonds between clay, GS, and CLS after 28 days of curing. The FTIR spectra of GS-CLS for treated clay were studied for the combinations M1CLS0.5, M2CLS0.5, and M3CLS0.5, as 0.5% yielded the best performance out of all combinations. Figure 14 illustrates the newly formed functional groups in the studied clay when it is treated with GS and CLS. An O-H bond is observed at a wavelength of 3700 cm^−1^, a C-H bond is observed at 1645 cm^−1^, which is the benzene ring formation from the CLS reaction, and at 1040 cm^−1^, a C-O bond is observed, which is due to phenolic bond formation with clay minerals and CLS. Si-O and Al-OH bonds are observed at 680 cm^−1^ and 785 cm^−1^ due to the interaction of clay, GS, and CLS.

### 3.3. Discussion of Reliability Analysis

#### 3.3.1. Effect of CP on β_CBR_ of GS- and CLS-Treated Soil

The reliability index (β_CBR_) against the CBR strength failure of the treated clay subgrade is affected by the curing period, as shown in Figure 15. In this figure, D_GS_ increases from 30% to 50% for a COV of D_GS_ = 5%, a COV of D_CLS_ = 5%, and a COV of CBR_min_ = 10% for CPs = 7 days, 14 days, and 28 days. Figure 16 shows that the reliability index (β_CBR_) is higher for the 28-day-cured sample than for the 14-day- and 7-day-cured samples. Figure 15 shows that the values of β_CBR_ are 0.4, 2.7, and 6.4 for CPs = 7, 14, and 28 days, respectively, when the clay soil is treated with GS and CLS. The reliability index (β_CBR_) increases by 5.75 and 15 times when the CP value is increased from 7 days to 14 days and 28 days, respectively. This indicates that the reliability of the low-volume roads constructed with the treated clay soil shows better performance when the clay-GS-CLS mix is cured for 28 days. 

#### 3.3.2. Effect of Dosage of Granite Sand (D_GS_) on Reliability Index (β_CBR_)

The results presented in Figure 16 show the effect of increasing the volume of granite sand (D_GS_) on the magnitude of β_CBR_. Figure 16 shows that at constant CP values, the magnitude of the reliability index significantly decreases as the amount of granite sand (GS) increases from 30% to 50%. This can likely be attributed to the reduction in the CBR value due to the reduction in basal bonding and the enhancement of peripheral bonding. Further, it appears that the granite sand content did not further enhance the CBR strength. This may be because the decrease in developed frictional resistance between the granite sand particles and clay reduced the developed tensile stresses in the clay-GS-CLS mixture.

#### 3.3.3. Effect of Dosage of Calcium Lignosulfonate (D_CLS_) on Reliability Index (β_CBR_)

The results presented in Figure 16 show the effect of increasing the volume of calcium lignosulfonate (D_CLS_) on the magnitude of the reliability index (β_CBR_). The results in Figure 16 indicate that calcium lignosulfonate (CLS) is a significant parameter that affects the behavior of the treated cohesive soil. Figure 16 reveals that when CP = 28 days, a marginal reduction in β_CBR_ from 6.37 to 6.10 is observed when the dosage of calcium lignosulfonate (CLS) increases from 0.5% to 2%. This can be attributed to the replacement of the soil with finer lignosulfonate.

#### 3.3.4. Optimal Dosages of Granite Sand (D_GS_) and Calcium Lignosulfonate (D_CLS_) for 28-Day Curing Period

Figure 17a–d present the optimal values of the dosage of granite sand (D_GS_) for various desired values of β_CBR_ for COVs of CBR_min_ = 10%, 20%, 30%, 40%, 50%, and 60% when the clay-GS-CLS mix is blended with dosages of calcium lignosulfonate (D_CLS_) = 0.5%, 1.0%, 1.5%, and 2.0%, respectively. It is noted from Figure 17a–d that the addition of 30 to 50% D_GS_ and 0.5% to 2.0% D_CLS_ reduces the value of β_CBR_. The magnitudes of reliability indices (β_CBR_) are 6.40, 3.35, 2.30, 1.80, 1.60, and 0.90 for COVs of CBR_min_ = 10%, 20%, 30%, 40%, 50%, and 60%, respectively, at D_GS_ = 30% when CP = 28 days and D_CLS_ = 0.5%. The addition of 1.0% CLS gives β_CBR_ of 6.28, 3.27, 2.29, 1.83, 1.57, and 1.41 for COVs of CBR_min_ = 10%, 20%, 30%, 40%, 50%, and 60%, respectively, at D_GS_ = 30%.

Figure 17a–d depict that maximum values of granite sand (D_GS_) = 35%, 34%, 33%, and 32% for D_CLS_ = 0.5%, 1.0%, 1.5%, and 2.0%, respectively, are needed to maintain the desired value of β_CBR_ ≥ 3.0 when the COV of CBR_min_ = 20% and CP = 28 days. However, Figure 17a–d show that when the COV of CBR_min_ > 20%, the addition of 30 to 50% D_GS_ and 0.5% to 2.0% D_CLS_ is inadequate to obtain the satisfactory performance of low-volume roads at β_CBR_ ≤ 3.0. 

### 3.4. Discussion of Carbon Footprint Analysis

#### 3.4.1. Detailed Description of the Stages Involved in the Estimation of CO_2_ Emissions for the Assumed Typical Pavement Subgrade

The subgrade of the pavement is assumed according to IRC SP 72-2007. The plan and C/S of the subgrade are depicted in Figure 18. The assumed subgrade has a 3.75 m width, 0.3 m depth, and 2 km length.

Stage I: Estimation of embodied carbon energy from materials. In this stage, the embodied carbon energy from the materials (clay, granite sand, calcium lignosulfonate, and water) is calculated based on the data established by Hammond and Jones [73] and Ashfaq et al. [41]. The quantities of the materials are calculated for subgrade pavement with a volume of 2250 m^3^, which affords a mass of 2.75 × 10^6^ kg clay and 1.179 × 10^6^ kg granite sand in order to achieve a density of 1.75 g/cc. Water with a measured volume of 640.59 m^3^ is used to achieve the required density. Table 4 shows the total embodied carbon emissions from the materials. It also reports the carbon analysis for Stage I.

It is inferred from Table 4 that the presence of GS as a replacement for clay reduced the overall carbon emissions due to the low carbon factor (0.0052) [83].

Stage II: Estimation of embodied carbon energy from the procurement and haulage process. In this stage, the carbon emissions obtained due to the procurement of materials by a pickup excavator with a 10 t/L capacity are considered. A heavy-duty dumper with a 25 t/L capacity is deployed for a haulage distance of 1 km. The embodied carbon factor (ECF) for this stage is based on the fuel on which the machine runs. The ECF of the fuel is sourced from Shillaber et al. [71]; Davis et al. [84]; and Kecojevic and Komljenoioc [85]. Table 5 shows the total embodied carbon emissions during the excavation of materials and the haulage distance along which the materials are transported. In Table 5, the embodied carbon emissions of the material are based on the capacity and type of the vehicle, the number of trips, the type of fuel used by that vehicle, and the haulage distance. Irrespective of the carbon factor of the material, the haulage distance influences the total embodied carbon of that particular operation.

Stage III: Estimation of total embodied carbon emissions during site operations. Site operations include spreading the materials (clay and granite sand), mixing calcium lignosulfonate using a mixer, spraying the chemical additive onto the surface of clay mixed with granite sand, and compacting the soil with a Smooth Wheel Roller. Table 6 describes the carbon emissions from the listed equipment during site operations, and the detailed carbon emissions during the construction of the subgrade are reported in Table 7.

#### 3.4.2. Comparison of Carbon Emissions of Different Stabilizers

A comparative study was performed to determine the carbon energy emitted when cement, lime, GS, and CLS are used as additives in the studied soil (intermediate compressible clay). The materials were quantified with the measured mass of clay used for the typical pavement section referred to in Figure 19. It is evident from Figure 19 that granite sand (GS) and calcium lignosulfonate (CLS) save 97.52% and 98.73% of carbon energy, respectively, at their optimal levels i.e., 70% clay and 30% GS, when compared to lime and cement.

The energy calculations for the traditional stabilizers are based on the optimal dosages of lime and cement, i.e., 6% lime and 4% cement, for an intermediate-plasticity clay according to Garzon et al. [86] and Prusinski and Sankar [87]. The embodied carbon factors for lime and cement are 0.76 and 0.95, respectively [73].

From Table 5, the carbon emissions are reported for a 1 km haulage distance in order to simplify the estimate for different and longer distances with respect to the site location. A comparative study was performed on the embodied carbon emissions of lime, cement, GS, and CLS at their optimal dosages for an intermediate-plasticity clay for a haulage distance of 7 km. This variation explores the effect of the haulage distance on the carbon emissions of the materials used in the field.

The embodied carbon energy of the materials observed in Figure 20 is contradictory to the existing criteria that the materials used in the study exhibit lower carbon emissions, which is mentioned in the above sections. This response is due to the haulage distance. In the current study, 30% GS is massive when compared to 6% lime, 4% cement, and 0.5% CLS in terms of the quantity of the material calculated. This increase in carbon emissions (74.28%) for GS is compensated by further mixing with the clay. As shown in Figure 20, 74.28% of carbon energy is saved by utilizing the GS from the massive dump near the industry.

## 4. Summary and Conclusions

The combined effect of granite sand (GS) and calcium lignosulfonate (CLS) on enhancing the performance of a clay subgrade for low-volume roads was studied. The California Bearing Ratio (CBR) was used as a performance indicator, and a curing period of 28 days was used for clay-GS-CLS mixes. The reliability-based design optimization of clay-GS-CLS mixes was carried out to estimate the optimal dosages of GS and CLS for the satisfactory performance of low-volume roads against CBR strength failure. Besides this, the stabilized clay soil was considered as a pavement material and examined for its carbon emissions during the construction of a pavement subgrade. The obtained carbon emissions were compared with traditional stabilizers to justify the usage of these sustainable stabilizers. The following conclusions are drawn from the present study:The addition of GS to the virgin soil at a constant volume reduces the CBR of the clay-GS matrix. The addition of CLS to the clay-GS mix enhances the clay-GS adhesion, resulting in higher CBR values of clay-GS-CLS mixes. At 0.5% CLS, the CBR values increased for the M1, M2, and M3 mixes, and the effect was more pronounced with an increase in the curing period. However, with a further increase in the CLS dosage up to 1.5%, the penetration resistance and CBR values were reduced, except at the 2% dosage.Strong and prominent chains are observed for 0.5% CLS in the presence of any dosage of GS due to the formation of chemical bonds. These are evidenced by micrograph images (SEM) and infrared spectra (FTIR). The reliability-based design optimization has revealed that the mean values of D_GS_ and D_CLS_ are the most sensitive random parameters that significantly influence the subgrade material stability of low-volume roads.The COV of the minimum specified value of the CBR considerably influences the stability of low-volume roads constructed with the clay soil blended with GS and CLS. It is demonstrated that the volumes of granite sand (D_GS_) and calcium lignosulfonate (D_CLS_) should be decreased for the desired performance of low-volume roads with an increase in the COV of CBR_min_ from 10 to 60%.The addition of 30 to 50% D_GS_ and 0.5% to 2.0% D_CLS_ is inadequate to obtain the desired performance of low-volume roads at β_CBR_ ≤ 3.0 in terms of the CBR strength when the COV of CBR_min_ is 30%.The embodied carbon emission factors of GS and CLS are 0.00526 and 0.2, respectively. These values are relatively low compared to conventional stabilizers such as lime (0.76) and cement (0.95). 

The carbon footprint analysis revealed that blending 30% GS and 0.5% CLS with clay yielded significant savings in terms of equivalent carbon emissions compared to the traditional stabilizers lime (6%) and cement (4%) at their respective optimal dosages. It is interesting to note that, irrespective of the material, the carbon emissions during Stage II (procurement and haulage) and Stage III (site operations) depend on the haulage distance and the type of fuel used by the equipment. The embodied carbon emissions of GS are reduced by 27 times compared to those of lime and 23 times compared to those of cement at their respective optimal dosages for a fixed haulage distance.

## Figures and Tables

**Figure 1 materials-16-02065-f001:**
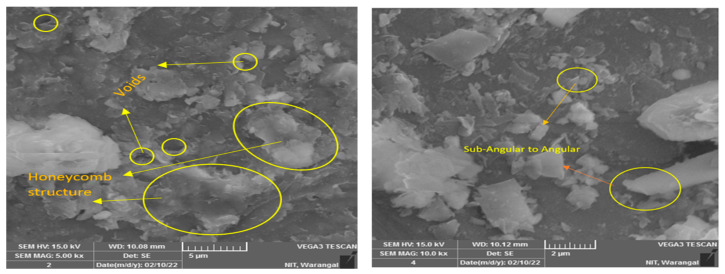
SEM images of clay and granite sand used in the study.

**Figure 2 materials-16-02065-f002:**
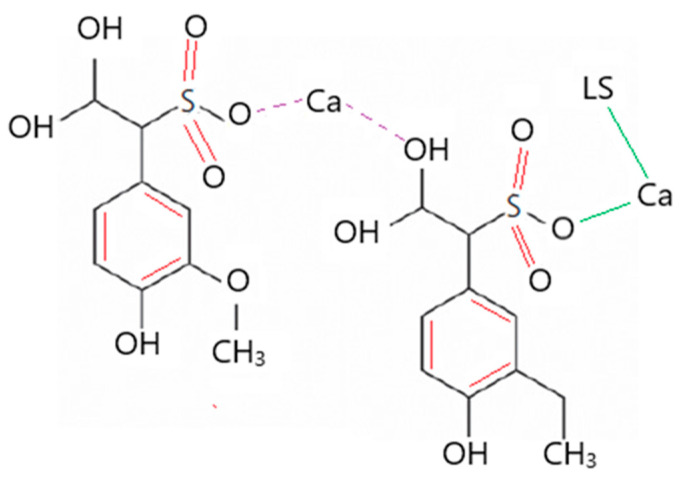
Chemical structure of CLS.

**Figure 3 materials-16-02065-f003:**
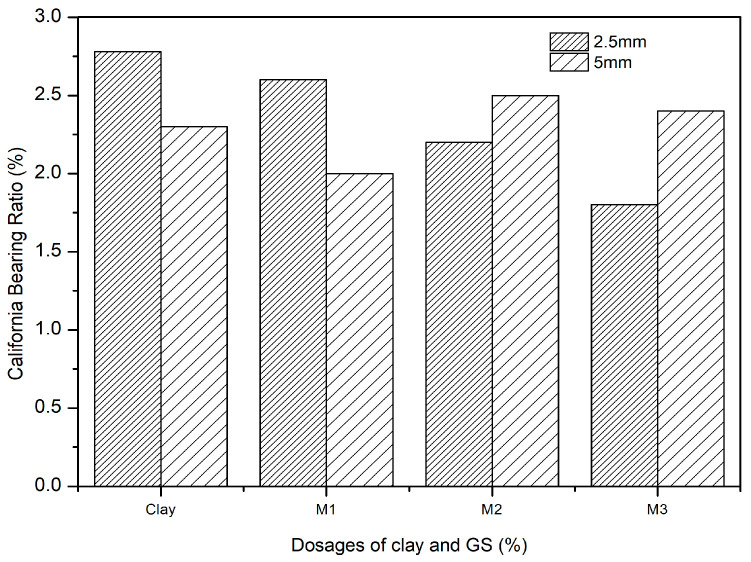
Variation in unsoaked CBR in the presence of GS.

**Figure 4 materials-16-02065-f004:**
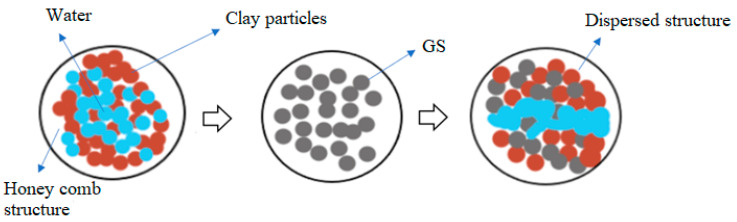
Change in soil structure with addition of GS at constant water content.

**Figure 5 materials-16-02065-f005:**
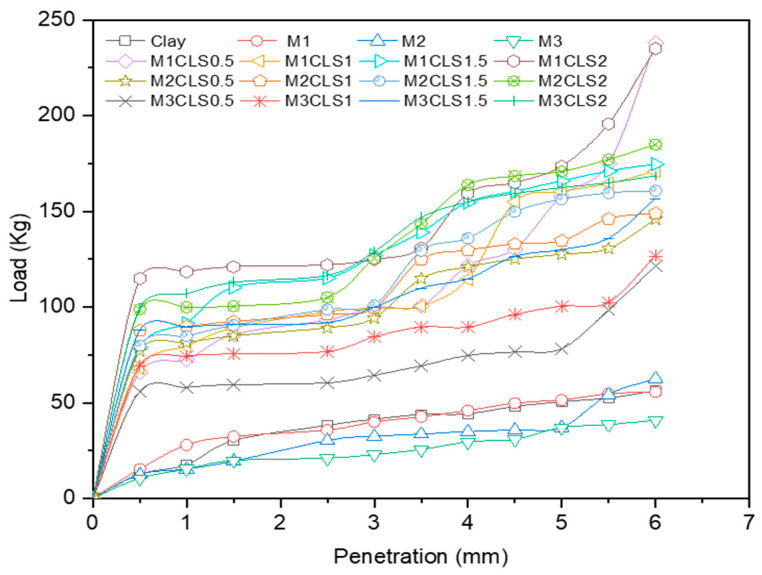
Load–penetration curves of clay-GS-CLS mix after 0 days of curing.

**Figure 6 materials-16-02065-f006:**
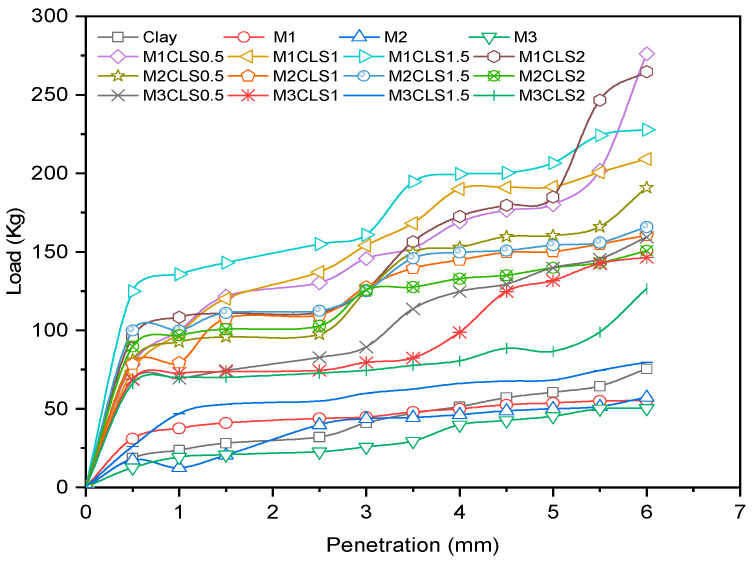
Load–penetration curves of clay-GS-CLS mix after 7 days of curing.

**Figure 7 materials-16-02065-f007:**
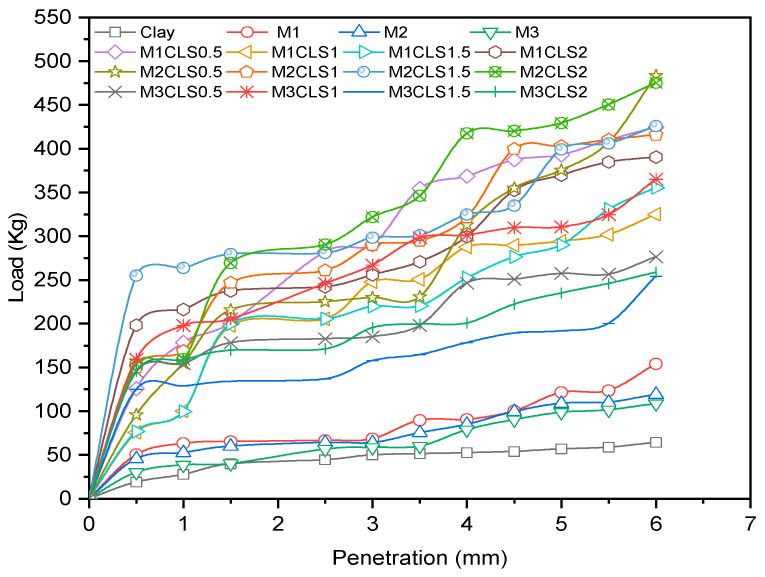
Load–penetration curves of clay-GS-CLS mix after 28 days curing.

**Figure 8 materials-16-02065-f008:**
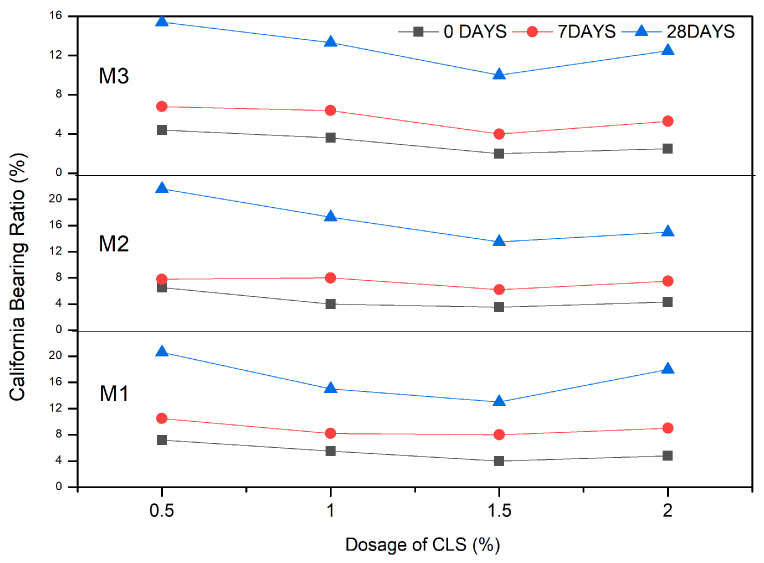
Variation in CBR of soil in presence of GS and CLS.

**Figure 9 materials-16-02065-f009:**
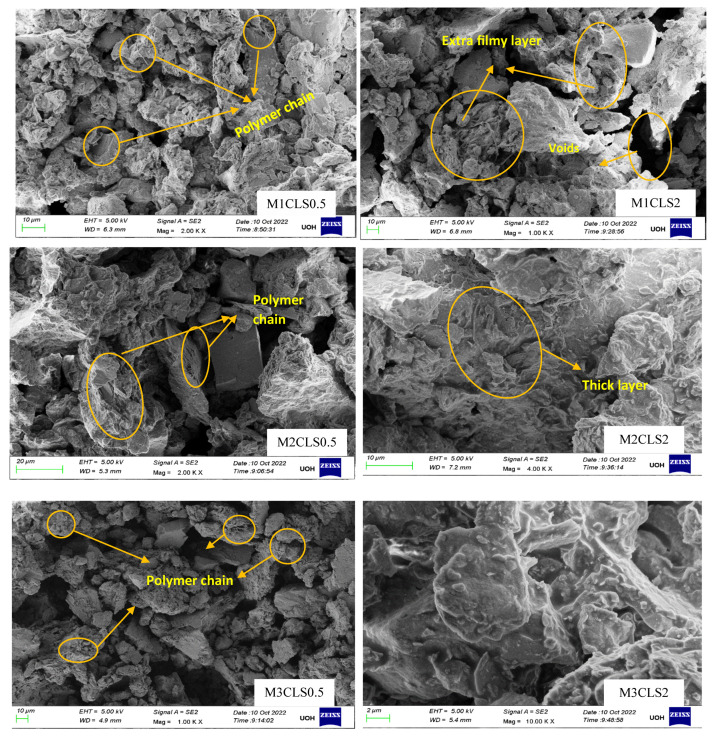
SEM images of binary blended clay after 28 days of curing.

**Figure 10 materials-16-02065-f010:**
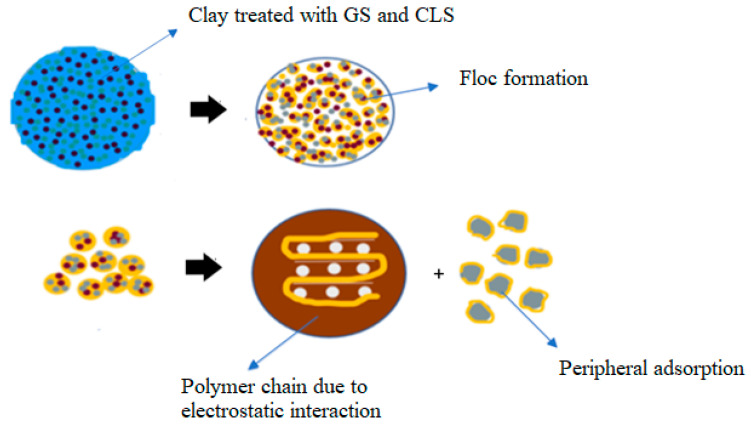
Interaction of CLS with clay and GS.

**Figure 11 materials-16-02065-f011:**
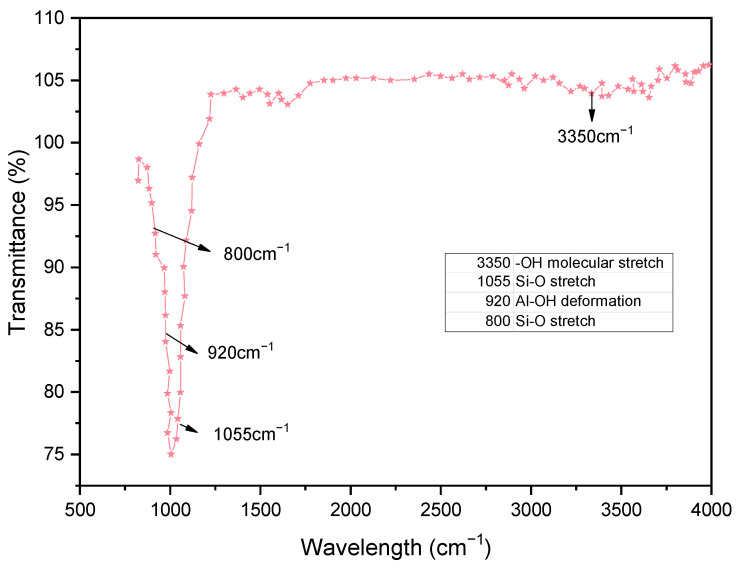
FTIR pattern of clay.

**Figure 12 materials-16-02065-f012:**
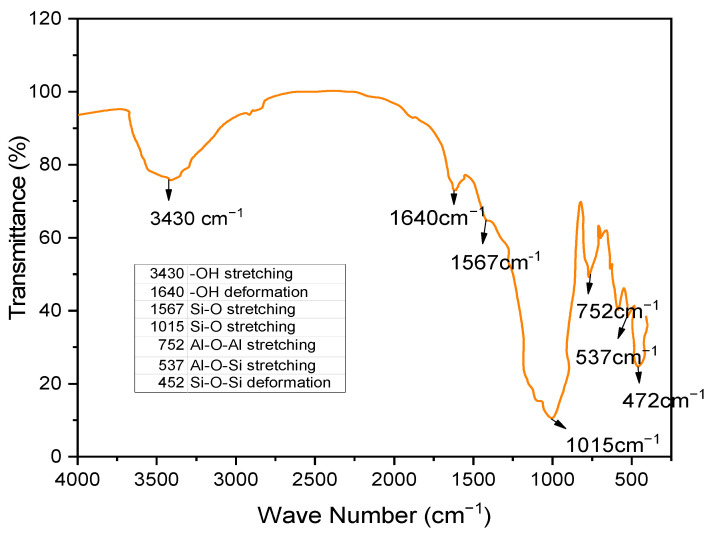
FTIR pattern of granite sand.

**Figure 13 materials-16-02065-f013:**
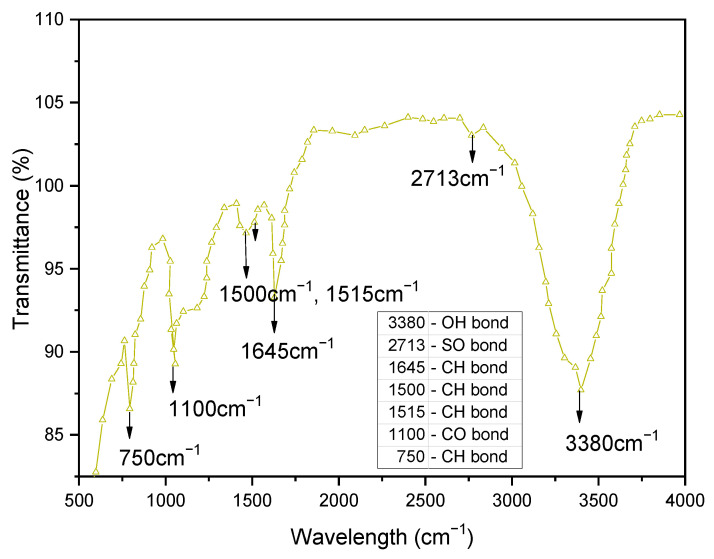
FTIR pattern of calcium lignosulfonate.

**Figure 14 materials-16-02065-f014:**
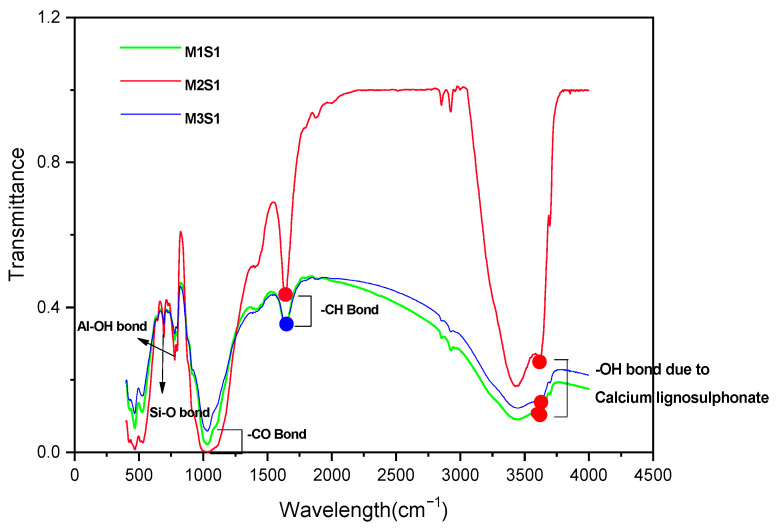
FTIR patterns of clay-GS mix in presence of 0.5% CLS.

**Figure 15 materials-16-02065-f015:**
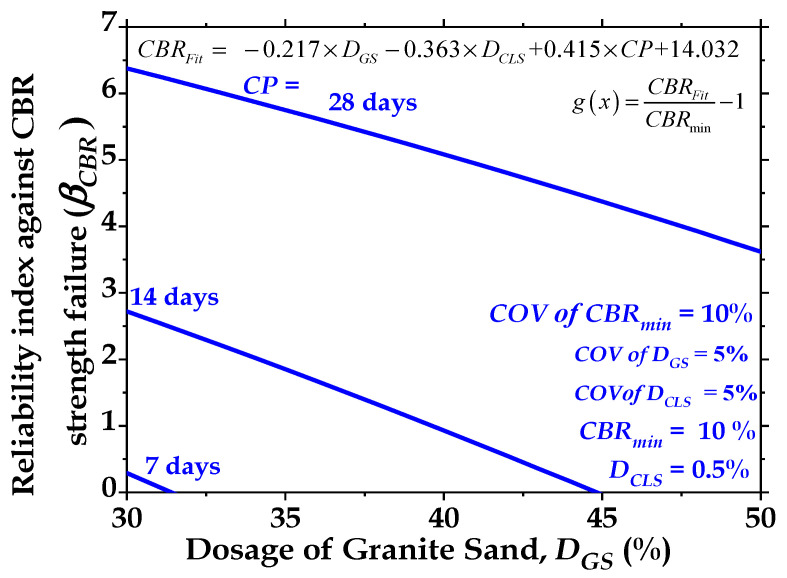
Effect of curing period (CP) on reliability index against CBR strength failure ($\beta_{\mathrm{CBR}}$) with dosages of granite sand ($D_{\mathrm{GS}}$) for dosage of CLS ($D_{\mathrm{CLS}}$) = 0.5%.

**Figure 16 materials-16-02065-f016:**
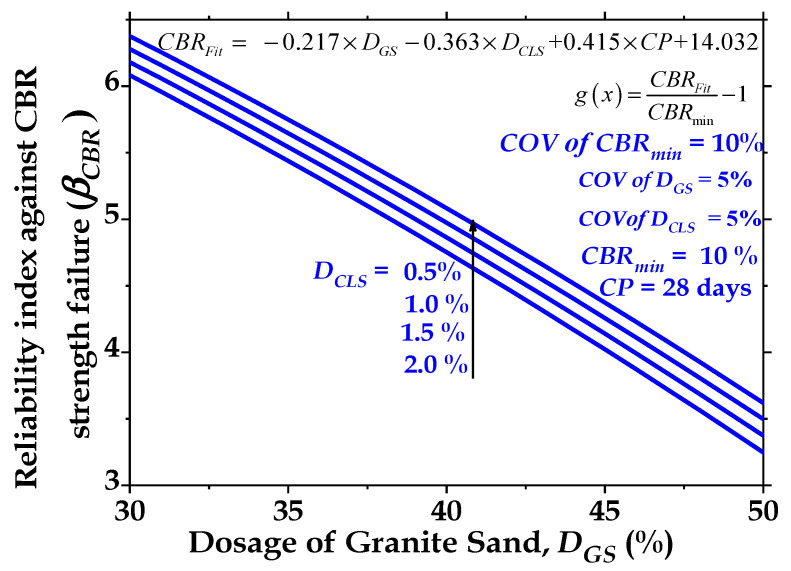
Effect of dosage of CLS ($D_{\mathrm{CLS}}$) on reliability index against CBR strength failure ($\beta_{\mathrm{CBR}}$) with dosages of granite sand ($D_{\mathrm{GS}}$) for curing period (CP) = 28 days.

**Figure 17 materials-16-02065-f017:**
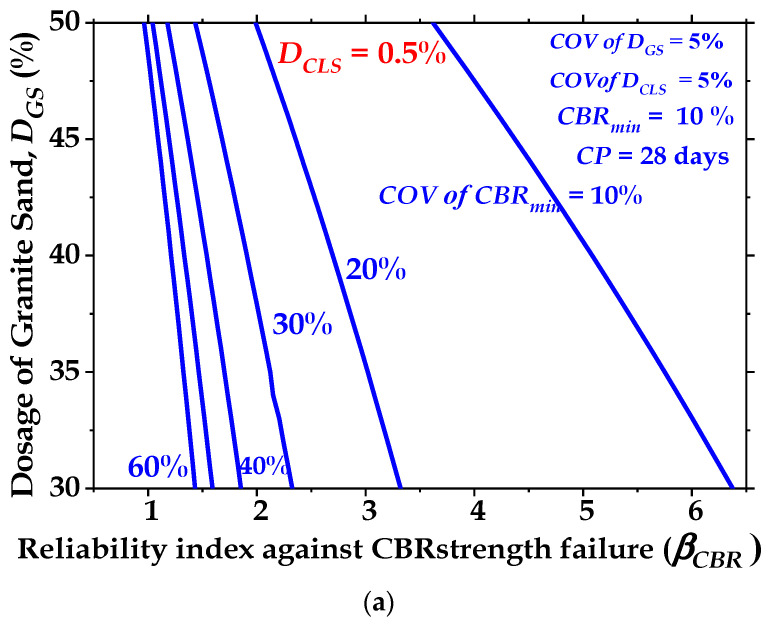
(**a**) Optimal values of dosages of granite sand ($D_{\mathrm{GS}}$) for various values of reliability index against CBR strength failure ($\beta_{\mathrm{CBR}}$) for 28-day-cured treated clayey soils when $D_{\mathrm{CLS}}$ = 0.5%; (**b**) optimal values of dosages of granite sand ($D_{\mathrm{GS}}$) for various values of reliability index against CBR strength failure ($\beta_{\mathrm{CBR}}$) for 28-day-cured treated clayey soils when $D_{\mathrm{CLS}}$ = 1.0%; (**c**) optimal values of dosages of granite sand ($D_{\mathrm{GS}}$) for various values of reliability index against CBR strength failure ($\beta_{\mathrm{CBR}}$) for 28-day-cured treated clayey soils when $D_{\mathrm{CLS}}$ = 1.5%; (**d**) optimal values of dosages of granite sand ($D_{\mathrm{GS}}$) for various values of reliability index against CBR strength failure ($\beta_{\mathrm{CBR}}$) for 28-day-cured treated clayey soils when $D_{\mathrm{CLS}}$ = 2.0%.

**Figure 18 materials-16-02065-f018:**
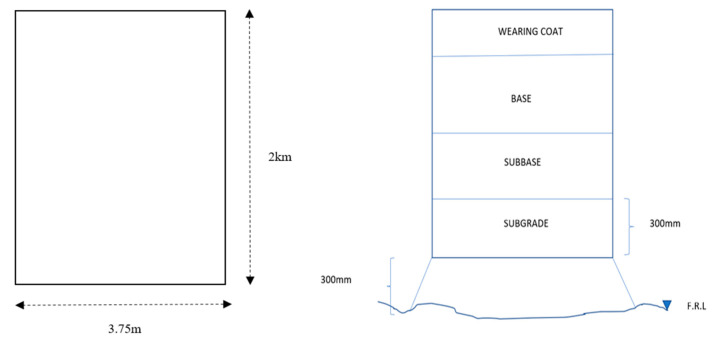
Typical cross-section of the assumed pavement subgrade.

**Figure 19 materials-16-02065-f019:**
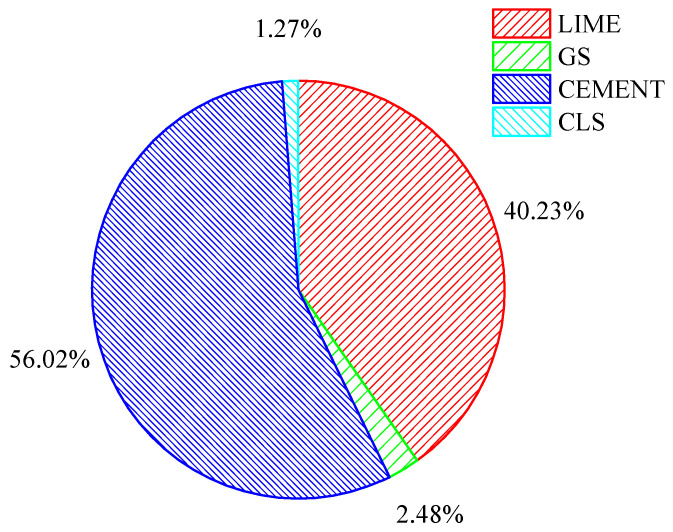
Comparative savings in carbon energy with traditional stabilizers.

**Figure 20 materials-16-02065-f020:**
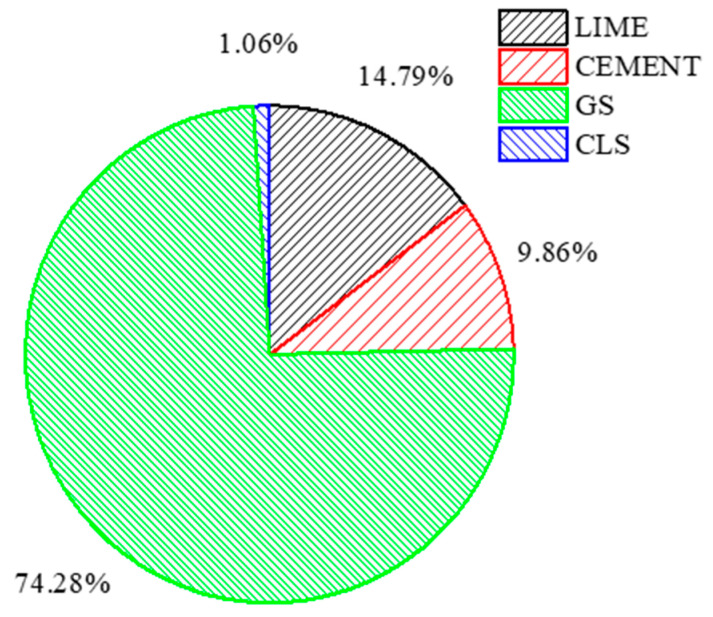
Variation in carbon emissions of materials for a haulage distance of 7 km.

**Table 1 materials-16-02065-t001:** Index properties and Chemical composition of Clay and GS.

Property	Value (Clay)	Value (GS)	Chemical Composition	Value of Clay (%)	Value of GS (%)
Specific gravity	2.62	2.72	Silica (SiO_2_)	55.34	53.06
Liquid limit (%)	45.13	-	Alumina (Al_2_O_3_)	9.92	6.16
Plastic limit (%)	22.34	-	Calcium Oxide (CaO)	1.06	1.64
Plasticity Index (%)	22.79	-	Magnesium Oxide (MgO)	1.97	5.86
Shrinkage limit (%)	13	-	Titanium Oxide (TiO_2_)	1.13	0.32
% Fines	63	10	Ferric Oxide (Fe_2_O_3_)	8.15	9.06
IS classification	CI	SP-SM	Sodium Oxide (Na_2_O)	0.31	1.37
DFS (%)	33	-			
Maximum Dry Density (g/cc)	1.75	2.1			
Optimum Moisture Content (%)	16.3	8.3			

**Table 2 materials-16-02065-t002:** The linear equation results for the California Bearing Ratio (${\mathrm{CBR}}_{\mathrm{fit}}$) of untreated clay measured after 0, 7, and 28 days of curing.

CP (Days)	CBR	$CBR_{fit}$	Residual	% Error
0	2.78	2.79	−0.01	−0.25
7	2.94	2.93	0.01	0.37
28	3.26	3.26	0	−0.12

**Table 3 materials-16-02065-t003:** The linear equation results for the California Bearing Ratio (${\mathrm{CBR}}_{\mathrm{fit}}$) of soil treated with granite sand (GS) and calcium lignosulfonate (CLS) measured after 7 and 28 days of curing.

CP (Days)	$D_{GS}$ (%)	$D_{CLS}$ (%)	CBR	$CBR_{fit}$	Residual	% Error
7	30	0.5	9.50	10.26	−0.76	−7.96
		1.0	10.00	10.07	−0.07	−0.74
		1.5	11.30	9.89	1.41	12.45
		2.0	9.00	9.71	−0.71	−7.90
	40	0.5	7.80	8.09	−0.29	−3.70
		1.0	8.00	7.91	0.09	1.16
		1.5	8.20	7.73	0.47	5.78
		2.0	7.50	7.54	−0.04	−0.59
	50	0.5	6.80	5.92	0.88	12.91
		1.0	6.40	5.74	0.66	10.30
		1.5	4.00	5.56	−1.56	−38.98
		2.0	5.30	5.38	−0.08	−1.46
28	30	0.5	20.60	18.97	1.63	7.89
		1.0	15.00	18.79	−3.79	−25.29
		1.5	15.00	18.61	−3.61	−24.08
		2.0	18.00	18.43	−0.43	−2.39
	40	0.5	17.30	16.81	0.49	2.84
		1.0	19.60	16.63	2.97	15.17
		1.5	20.50	16.45	4.05	19.78
		2.0	21.20	16.26	4.94	23.28
	50	0.5	13.33	14.64	−1.31	−9.84
		1.0	15.40	14.46	0.94	6.11
		1.5	10.00	14.28	−4.28	−42.78
		2.0	12.50	14.10	−1.60	−12.77

**Table 4 materials-16-02065-t004:** Evaluation of carbon emissions from materials.

Stage I	Material (1)	Amount (m^3^) (2)	Unit Weight (kg/m^3^) (3)	Weight (t) (4)	ECF (5)	Embodied Carbon (t) CO_2_e/t (6) = (4) × (5)
Embodied carbon of the material	Clay (CI)	2250	1750	2.75 × 10^3^	0.0056	15.4
	GS	2250	1750	1.179 × 10^3^	0.0052	6.13
	CLS	2250	-	19.65	0.2	3.93
	Water	6405.9	1000	0.6405 × 10^3^	0.001	0.64
Total CO_2_(t) emissions in Stage I = **26.1**						

**Table 5 materials-16-02065-t005:** Evaluation of carbon emissions from procurement and haulage.

Stage II	Process	Vehicle	Capacity (t/L)		No. of Loadings	Total Fuel (L)	ECF (Fuel Based Equipment)	Embodied Carbon (t) CO_2_e/t
Excavation and Procurement	Clay procurement	Pickup excavator	10		275	275	3.25	893.75
	GS procurement	Pickup excavator	10		118	118	3.25	383.5
	CLS	Pickup excavator	10		2	2	3.25	6.5
Total CO_2_								**1283.75**
**Haulage**	**Process**	**Vehicle**	**Capacity (t/L)**	**Distance** **(km)**	**Trips**	**Total fuel** **(L)**		**Embodied carbon** **CO_2_e/t (t)**
Haulage	Clay	Heavy-duty dumper	25	1	55	55	3.25	178.75
	Granite sand	Heavy-duty dumper	25	1	24	23.58	3.25	76.635
	Calcium lignosulfonate	Heavy-duty dumper	25	1	0.4	0.4	3.25	1.277
Total CO_2_								**256.66**
Total CO_2_(t) emissions in Stage II = **1540.4**								

**Table 6 materials-16-02065-t006:** Evaluation of carbon emissions from site operations.

Stage III	Process	Vehicle/Machine	Capacity	No. of Trips	Total Fuel (L)	ECF	Embodied Carbon (t) CO_2_e/t
Site operation	Spreading	Bulldozer	10 t/L	393	393	3.25	1276.9
Haulage	Mixing of CLS	Slurry mixer	0.5 t (50 lb)	40	40	3.25	127.7
	Spraying of CLS	Distributor truck	500 L	1.3	1.3	3.25	4.25
	Compaction	Smooth Wheel Roller	12 t/L	328	328	3.25	1064.4
Total CO_2_							**3622.7**
Total CO_2_(t) emissions in Stage III = **3622.7**							

**Table 7 materials-16-02065-t007:** Summations of stage wise carbon emissions with GS and CLS as a pavement application.

Stage	Operation	Embodied Carbon (CO_2_e/t)
Stage I	Material	**26.1**
Stage II	Haulage	**1283.75**
	Procurement	**256.66**
Stage III	Site operations	**3622.7**

## Data Availability

Not applicable.

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
