# Peer review of "Sustainable Binary Blending for Low-Volume Roads—Reliability-Based Design Approach and Carbon Footprint Analysis"

_materials, 2023, doi:10.3390/ma16052065_

Round 1

Reviewer 1 Report

1. In Fig. 1, whether there is SEM image of Calcium Lignosulphonate (CLS)?

2. Contents in Table 1 are confusing, please re-arrange the table.

3.Section 2.6, the definition of Low Volume Roads (LVR) is missing.

4. Line 226, the Fig. 6 appears before Fig. 3.

5. In Fig. (9), the scale bars of SEM images are missing.

6. The conclusion part is too long and should be more concluded and concise.

7. In the conclusion section, whether the design approach for the binary blending of CLS and GS is just for the subgrade of the LVR, in which the definition or an illustration figure is missing.

Author Response

The authors would like to thank the reviewers for their constructive comments which helped the cause of the manuscript. The authors have addressed all the comments raised by the reviewer and highlighted the suggestions using “Red Colored Text” in the revised manuscript.

Query 1: In Fig. 1, whether there is SEM image of Calcium Lignosulphonate (CLS)?

Response: Clarification. The SEM analysis of CLS was unable to conducted due to it’s hydrophilic nature. Hence, the Fig. 1 in the manuscript represented only the SEM images of clay and GS as mentioned in section 2.1.

Query 2: Contents in Table 1 are confusing, please re-arrange the table.

Response: Agreed and revised as suggested. The authors have incorporated the suggestion in the revised manuscript and rearranged the Table 1.

Query 3:  Section 2.6, the definition of Low Volume Roads (LVR) is missing.

Response: The authors thank the reviewer for pointing this mistake due to oversight. The relevant section (Section 2.6, Lines 208-210) was revised in the manuscript with a suitable definition.

Query 4: Line 226, the Fig. 6 appears before Fig. 3.

Response: Apologies. The authors have revised the Line 226 with a supporting phrase given in the Line 228.

Query 5: In Fig. (9), the scale bars of SEM images are missing.

Response: Clarification. The scale bars of SEM images were cropped during labelling and formatting the figure. Accordingly, these figures have been retained in the revised manuscript.

Query 6: The conclusion part is too long and should be more concluded and concise.

Response: Clarification. Each conclusion is drafted to represent both qualitative and quantitative research findings of the study and they couldn’t be more reduced further. Accordingly, they have been retained.

Query 7: In the conclusion section, whether the design approach for the binary blending of CLS and GS is just for the subgrade of the LVR, in which the definition or an illustration figure is missing.

Response: Clarification. In section 3.4, Figure 18 represents the practical application of the binary blending of GS and CLS. Figure 18 comprises the plan view of the subgrade of LVR section given in Figure 18 (Plan of LVR subgrade and cross section of LVR).

Reviewer 2 Report

The presented article is very clear and cleanly written. The article uses both metric and English units of measurement. I understand that the use of different units is due to different measurement approaches. If this could be unified, ideally into metric units, the resulting clarity effect would be more pronounced.

You are using units of volumetric mass g/cc. Wouldn't it be more appropriate to use the base SI units of kg/m³ (eg line 402) if you are using the derived units of kPa (eg Table 3). However, this is a small thing that is understandable.

I would like to point out the appropriate use of lower and upper indexes CO2 × CO2, M3 × m³, etc.

Author Response

The authors would like to thank the reviewers for their constructive comments which helped the cause of the manuscript. The authors have addressed all the comments raised by the reviewer and highlighted the suggestions using “Red Colored Text” in the revised manuscript.

Query 1: The presented article is very clear and cleanly written. The article uses both metric and English units of measurement. I understand that the use of different units is due to different measurement approaches. If this could be unified, ideally into metric units, the resulting clarity effect would be more pronounced.

Response: The authors thank the reviewer for positive feedback in this aspect. The usage of metrics is in accordance with relevant ASTM standards and IS standards when dealing with geotechnical properties. Whereas when it comes to application part, English units have to be used as they are the norm.

Query 2: You are using units of volumetric mass g/cc. Wouldn't it be more appropriate to use the base SI units of kg/m³ (eg line 402) if you are using the derived units of kPa (eg Table 3). However, this is a small thing that is understandable.

Response: Usually the properties of soil are denoted in volumetric mass (g/cc) and SI units are avoided. kPa being a pressure unit, it has been retained in accordance with ASTM codal norms.

Query 3: I would like to point out the appropriate use of lower and upper indexes CO2 × CO2, M3 × m³, etc.

Response: Apologies for the mistake. In section 2.8, the term CO2 has been corrected at relevant places and in the running text.

Clarification for M3 – In this manuscript M3 denotes, 50% clay and 50% GS mixture adopted which has been clearly mentioned in Section 2.3. Accordingly, it has been retained in the revised version of manuscript.

Reviewer 3 Report

Review Paper ‘Sustainable Binary blending for Low Volume Roads - Reliabil- 2 ity-Based Design Approach and Carbon Footprint Analysis’

In this paper , Granite sand (GS) and Calcium lignosulphonate (CLS) 14 are used as alternatives to traditional stabilizers for cohesive soil (clay). Interesting results are given but their presentation is not good. The reading of the manuscript is very complicated. Too many figures and little synthesis.

Q1 .GS was subjected to the X-ray generated by the Scanning Electron Microscope to study the fabric of the particle. Figure. 1 is a micrograph obtained from SEM analysis which depicts the particles of GS 146 are angular-sub angular, flaky, and completely granular.

The vocabulary used by the other must be corrected. SEM technique generates electron beam and not X-rays.

Q2- uncertainty about values of the chemical results given in table 1 must be indicated.

Q3- The figure 6 is announced before figure 3. Why ?

Q4- The paragraph 2.7.1. Need for Reliability Based Design, seems to introduction part and needs to be shifted as bibliographic part.

Q5- line 290 :  treated clay soils as random instead of treated clayey soils as random.

Q6- FTIR spectra should be drawn in solid line only. Sometimes it is not possible to distinguish the details.

Q7- Sometimes the figures are placed very far from the description which makes reading the text very difficult.

Q8- In figure 9 SEM image identification of polymer chains needs confirmation ( for example by X-ray microanalysis spectrometry).

Author Response

The authors would like to thank the reviewers for their constructive comments which helped the cause of the manuscript. The authors have addressed all the comments raised by the reviewer and highlighted the suggestions using “Red Colored Text” in the revised manuscript.

Query 1: GS was subjected to the X-ray generated by the Scanning Electron Microscope to study the fabric of the particle. Figure. 1 is a micrograph obtained from SEM analysis which depicts the particles of GS 146 are angular-sub angular, flaky, and completely granular. The vocabulary used by the other must be corrected. SEM technique generates electron beam and not X-rays.

Response: Agreed and revised as suggested in the Line 145 of Section 2.2. The authors have corrected the term to ‘electron beam’ in the revised manuscript.

Query 2: uncertainty about values of the chemical results given in table 1 must be indicated.

Response: Agreed and revised as suggested. The authors have incorporated the suggestion in the revised manuscript and rearranged the Table 1.

Query 3: The figure 6 is announced before figure 3. Why?

Response: Apologies. The authors have revised the manuscript with a supporting phrase given in the Line 228. The figure numbers are properly cited in the revised manuscript.

Query 4:  The paragraph 2.7.1. Need for Reliability Based Design, seems to introduction part and needs to be shifted as bibliographic part.

Response: Clarification. The paragraph 2.7.1 – Need for Reliability Based Design has to be explained before the analysis. Accordingly, it cannot be moved to bibliographic part and has been retained in the revised version of manuscript.

Query 5: line 290:  treated clay soils as random instead of treated clayey soils as random.

Response: The authors thank the reviewer for pointing this mistake due to oversight. The relevant correction in Line 291 of section 2.7.2 has been incorporated in the revised manuscript.

Query 6: FTIR spectra should be drawn in solid line only. Sometimes it is not possible to distinguish the details.

Response: Agreed and revised as suggested in the revised manuscript from Fig. 11 through Fig. 14. The authors thank the reviewer for this constructive criticism.

Query 7: Sometimes the figures are placed very far from the description which makes reading the text very difficult.

Response: Apologies. The manuscript was formatted in order to include all relevant sections at appropriate places. Hence, at some places the figures appear to be placed away from the cited text.

Query 8: In figure 9 SEM image identification of polymer chains needs confirmation (for example by X-ray microanalysis spectrometry).

Response: Clarification. The FTIR spectra of different combinations mentioned in the Fig. 14 confirms the formation of the polymer chain highlighted in Fig. 9. Accordingly, X-ray microanalysis spectrometry has not been carried out in the study due to limited resources available.

Reviewer 4 Report

Sustainable Binary blending for Low Volume Roads - Reliability-Based Design Approach and Carbon Footprint Analysis has been reviewed with the following comments;

* The introduction and the reviewed literatures are sufficient.

* The geographical map of Battupally 129 lake, Telangana, India (17.97370N 79.53520E) needs inclusion in the appropriate section to strengthen the research paper's technical content and assist future research works on this area.

* The results are well and extensively presented. 

* The conclusions are supported by the results of this research. 

Author Response

The authors would like to thank the reviewers for their constructive comments which helped the cause of the manuscript. The authors have addressed all the comments raised by the reviewer and highlighted the suggestions using “Red Colored Text” in the revised manuscript.

Comment 1: The introduction and the reviewed literatures are sufficient.

Response: The authors thank the reviewer for giving the positive remark.

Comment 2: The geographical map of Battupally 129 lake, Telangana, India (17.97370N 79.53520E) needs inclusion in the appropriate section to strengthen the research paper's technical content and assist future research works on this area.

Response: The authors thank the reviewer for extending the view to the future scope. But, the location mentioned in the manuscript is turned to a built-up area since the time of sampling (December 2019). The authors infer that geographical coordinates are equally reliable to the maps to unveil the location.

Comment 3: The results are well and extensively presented.

Response: The authors thank the reviewer for giving the positive remark.

Comment 4:  The conclusions are supported by the results of this research.

Response: The authors thank the reviewer for giving the positive remark.

Round 2

Reviewer 3 Report

The authors will have to be attentive in the future study to the synthetic aspect.

Author Response

The authors would like to thank the reviewers for their constructive comments which helped the cause of the manuscript. The following is our response to the query raised by the reviewer.

Comment 1: The authors will have to be attentive in the future study to the synthetic aspect.

Response: The main aim of the study is to exploit the bulk utilization of the targeted materials (granite sand and calcium lignosulphonate) from sustainable perspective. The authors would consider the suggestion raised by the reviewer while addressing pertinent synthetic applications using these targeted materials.
